# Assessment of the Arabic patient-centered online information about orthodontic pain: A quality and readability assessment

**Muath Saad Alassaf** [1]*, **Hatem Hazzaa Hamadallah** [2], **Abdulrahman Almuzaini**[2], **Aseel M. Aloufi**[2], **Khalid N. Al-Turki**[2], **Ahmed S. Khoshhal**[3], **Mahmoud A. Alsulaimani**[1], **Rawah Eshky**[1]

1 Department of Orthodontics and Dentofacial Orthopedics, College of Dentistry, Taibah University, Madina, Saudi Arabia, 2 College of Dentistry, Taibah University, Madina, Saudi Arabia, 3 Department of Restorative Dentistry, Dental Center in Ohud Hospital, Madina, Saudi Arabia

* Mo3ath345@gmail.com

## Abstract

### Background

This study assesses the quality and readability of Arabic online information about orthodontic pain. With the increasing reliance on the internet for health information, especially among Arabic speakers, it's critical to ensure the accuracy and comprehensiveness of available content. Our methodology involved a systematic search using the Arabic term for (Orthodontic Pain) in Google, Bing, and Yahoo. This search yielded 193,856 results, from which 74 websites were selected based on predefined criteria, excluding duplicates, scientific papers, and non-Arabic content.

### Materials and methods

For quality assessment, we used the DISCERN instrument, the Journal of the American Medical Association (JAMA) benchmarks, and the Health on the Net (HON) code. Readability was evaluated using the Simplified Measure of Gobbledygook (SMOG), Flesch Reading Ease Score (FRES), and Flesch-Kincaid Grade Level (FKGL) scores.

### Results

Results indicated that none of the websites received the HONcode seal. The DISCERN assessment showed median total scores of 14.96 (± 5.65), with low overall quality ratings. In JAMA benchmarks, currency was the most achieved aspect, observed in 45 websites (60.81%), but none met all four criteria simultaneously. Readability scores suggested that the content was generally understandable, with a median FKGL score of 6.98 and a median SMOG score of 3.98, indicating middle school-level readability.

### Conclusion

This study reveals a significant gap in the quality of Arabic online resources on orthodontic pain, highlighting the need for improved standards and reliability. Most websites failed to

**Data Availability Statement:** All relevant data are within the manuscript and its Supporting Information files.

**Funding:** The author(s) received no specific funding for this work.

**Competing interests:** The authors have declared that no competing interests exist.

**Abbreviations:** JAMA, Journal of the American Medical Association; HON, Health on the Net; SMOG, Simplified Measure of Gobbledygook; FRES, Flesch Reading Ease Score; FKGL, Flesch-Kincaid Grade Level.

meet established quality criteria, underscoring the necessity for more accurate and trustworthy health information for Arabic-speaking patients.

## Introduction

Pain is described as an unpleasant feeling sensed by the body and mind and emotional experience [1]. This feeling is linked to or characterized in relation to existing or possibly physiological tissue damage [1]. Pain is a subjective experience that differs widely between individuals. It is based on factors such as age, gender, an individual's discomfort threshold, the intensity of applied pressure, present sentiments and stress levels, cultural background, and previous pain experiences; this emphasizes pain as a complicated sensation impacted by both physical and psychological factors [2].

The rising prevalence of jaw and teeth irregularities combined with people's interest in enhancing their facial features through orthodontics has led to growing patient demand for corrective dental care [3]. More individuals are now seeking orthodontic treatment to address dental issues and improve their smiles [3]. During the course of orthodontic treatment, patients wearing orthodontic appliances feel pain and discomfort to varying degrees. One of the primary reasons individuals refuse orthodontic treatments is pain. Therefore, it is essential for both patients and orthodontists to manage pain [4]. Orthodontic treatment can often cause pain and discomfort, with studies reporting that up to 95% of orthodontic patients experience these symptoms at various stages of treatment [5, 6]. The pain can arise from procedures such as early interceptive extractions, placement of separators, bands, and archwires, and even during debonding and retainer fitting [5, 7, 8]. This pain not only affects the physical well-being of patients but also has a significant impact on their overall quality of life.

Pain and discomfort are common during active orthodontic treatment, especially with fixed appliances; the pain typically begins within four hours of appliance placement, peaks, and the pain typically begins within four hours of appliance placement, peaks, and subsides within seven days [9]. The main causes of pain and discomfort are pressure, ischemia, inflammation, and edema related to tooth movement [10]. Moreover, the type of orthodontic appliance used also influences pain levels. Studies show patients treated with fixed braces generally report significantly more intense pain versus those with removable aligners [11]. Invisalign clear aligners have demonstrated advantages, such as causing less discomfort compared to fixed appliances, especially in early treatment stages [12]. Several studies have been conducted to determine which areas of the mouth produce the greatest pain or discomfort during orthodontic treatment. According to the study's findings, among all teeth, the lower incisors frequently feel the greatest discomfort, both at initial alignment when braces are placed as well as later during debonding when braces are removed [13].

Pain management during orthodontic treatment is a significant area of investigation. Several studies have been conducted to investigate the usage of analgesics during procedures such as separator and archwire placement [10, 14]. The majority of the studies have been focused on ibuprofen and paracetamol [15]. Orthodontic pain treatment is a topic that researchers are examining using current technology, and laser technology is one of these ways that has had positive outcomes [16].

Furthermore, a study conducted on orthodontic patients found that orthodontists tend to underestimate the amount of Pain and discomfort their procedures induce [17]. Therefore, to improve how to manage pain and discomfort, orthodontists must gain an understanding into

the pain produced by orthodontic procedures. Orthodontists can better help patients by improving pain management and enhancing acceptance of required treatment stages with a more educated viewpoint [18]. This results in greater adherence to the intended procedure and more satisfaction with overall outcomes. A thorough understanding of orthodontic discomfort allows orthodontists to provide better care [4].

The internet has revolutionized access to health information, becoming a worldwide platform utilized by both healthcare professionals and non-experts [19]. With just a few clicks, individuals can access a wealth of health-related information that influences their decision-making processes and overall satisfaction with healthcare [19, 20]. However, it is important to acknowledge that online information can vary in quality, leading to potential misconceptions [20].

Given the internet's significant impact, it is reasonable to expect that individuals undergoing or interested in orthodontic treatment will seek online information about the associated pain. However, the unregulated nature of the web, combined with limited understanding of authoritative sources, increases the likelihood of encountering questionable information. Previous studies have examined web-based information in orthodontics, focusing primarily on extraction treatment, cleft lip and palate, and orthognathic surgery. These studies have consistently found variations in the quality of online information [19].

Furthermore, the increasing accessibility of the internet has led to a growing reliance on online health information among Arabic speakers [21]. However, concerns arise regarding the reliability of Arabic language health information online, including inaccuracies in translation, cultural misconceptions, and lack of accreditation [22]. Given that Arabic is the fifth most widely spoken language globally, it is essential to ensure the accuracy and comprehensiveness of health information available in Arabic [23].

Assessing online Arabic resources regarding pain during orthodontic treatment is crucial, given the reliance on the internet for health information. Conducting further studies that evaluate Arabic language sources, including websites and social media platforms, will help bridge the research gap and provide accurate information to individuals. By implementing this measure, Arabic-speaking patients can have reliable and comprehensive sources when researching pain during orthodontic treatment options online.

To date, no studies have assessed the quality and readability of Arabic health websites specifically focused on pain during orthodontic treatment. Therefore, the aim of our study is to examine the qualitative characteristics of publicly available electronic information regarding pain associated with orthodontic interventions.

## Materials and methods

### Website research and selection

In this retrospective analysis of the websites, to identify appropriate informants, the Arabic term "Orthodontic pain" as "ألم تقويم الأسنان" was typed into Google, Bing, and Yahoo. To eliminate various sorts of websites from the findings, specific criteria were developed. Unrelated information, duplicate pages, scientific/academic papers, multimedia content, social media, paid/sponsored links, advertisements, non-Arabic websites, and websites needing login or payment access were all filtered away. We were able to collect and analyze appropriate orthodontic pain materials available to the public in their preferred languages from search engines because of our methodical approach to acquiring internet search data based on pre-defined standards. This approach allowed for the collection and analysis of only pertinent information on orthodontic pain that was easily accessible through standard, unrestricted searches. The authors used incognito mode to prevent biases from influencing their search results and erased all search history and cookies before conducting their searches in October 2023.

Then, we categorized the websites we had into different groups based on how they were related to orthodontic pain. We also looked at where they came from (like if they were from a company, a non-profit group, a university, or a government), what kind of information they had (like medical facts, clinical trials, questions, and answers, or personal stories), and how they presented the information (using pictures, written text, or videos). This categorization was done using the categories proposed by Ni Riordain and McCreary [24].

## Quality assessment

The websites' quality and reliability were assessed using three different indicators: the Journal of the American Medical Association (JAMA) benchmarks for website analysis [25], the Health on the Net (HON) evaluation [26], and the DISCERN evaluation instrument [27].

These tools are established and widely recognized for evaluating the credibility and quality of health-related content. Our decision not to conduct a separate validation process for these tools in the context of Arabic texts was informed by precedent. Previous research has applied these evaluation tools across a variety of languages without specific validation for each linguistic context. This approach is based on the assumption that the criteria these tools assess—such as accuracy, transparency, and evidence-based content—are universal in nature and not dependent on language. Consequently, we proceeded with their application to Arabic texts under the same premise, focusing our efforts on the assessment of content quality and reliability.

The JAMA markers, introduced by the American Medical Association in 1997, consist of four key components. These components include disclosure, which involves openly revealing ownership, funding sources, donations, and conflicts of interest. Currency refers to the timeliness of the information, including dates of content sharing, updates, and revisions. Authorship recognizes the individuals responsible for the content, such as researchers and contributors. Lastly, attribution involves clearly indicating the sources of the information.

The accuracy of health information was evaluated using the DISCERN instrument, a standardized tool. This instrument comprises three categories and sixteen items, where each item is scored on a scale of 1 to 5, ranging from full rejection to complete approval respectively. The DISCERN instrument assesses validity through items 1 to 8 and 9 to 15, and the final item provides a general assessment. The total points obtained are then classified into three levels of quality: low (16–32), moderate (33–64), and high (65).

Furthermore, the Health on the Net (HON) code serves as a reputable indicator for credible online health information. Health-related websites can obtain HON code accreditation by meeting eight standards that ensure excellent quality and transparently disclosed data. Upon meeting these standards, a website is awarded a HON seal for one year, with annual evaluations required to maintain the accreditation.

The reliability of websites was assessed independently by two investigators who reviewed each site separately. To ensure unbiased results, their individual scores were averaged and used in the analysis.

## Readability evaluation

The validation of readability assessment tools for non-English content, it is important to note that the Gunning Fog Index (GFI), Coleman-Liau Index (CLI), and Automated Readability Index (ARI) were excluded from our analysis. The rationale behind this decision is rooted in the structural differences between English and Arabic languages. Specifically, these indices calculate readability scores based on the number of letters per word, a method not suitable for Arabic. In Arabic, words are not merely sequences of letters but are composed of morphemes

and other components that are interconnected, making the application of letter-count-based formulas inappropriate. For assessing readability in Arabic, we adhered to established benchmarks: a Flesch Reading Ease (FRE) score of 80.0 or higher and Flesch-Kincaid Grade Level (FKGL) and Simple Measure of Gobbledygook (SMOG) scores below 7, aligning with precedents in previous Arabic studies. This approach ensures comparability and appropriates application of readability assessment in Arabic texts. [28–30].

## Statistical analysis

Quantitative data are presented as median with interquartile range according to the result of the Kolmogorov-Smirnov test.

## Result

### Available websites

While exploring the Arabic term for "orthodontic pain" on various search engines, Google yielded 123,056 websites, Bing produced 25,800, and Yahoo presented 45,000 results. The assessment involved the first 100 results from each search engine against eligibility criteria, leading to the identification of 74 websites that met the specified requirements. The exclusion of the remaining 226 websites as explained in **Fig 1**, providing a detailed overview of our selection strategy.

An analysis of websites revealed that the majority (73%) were operated by non-profit organizations. Commercial websites represented the next largest category (23%), while universities and medical centers combined accounted for only 4%. Notably, none of the websites were affiliated with the government.

In terms of specialization, all websites were partly related to the subject matter. Medical facts were the most prevalent content type, appearing on 96% of websites. Q&A and clinical trial information were rare, each only found on one website. Visual content was also limited: images appeared on 26% of websites and videos on only 3%. Audio was entirely absent.

**Table 1** provides a detailed breakdown of the website categories according to affiliation, specialization, content type, and presentation.

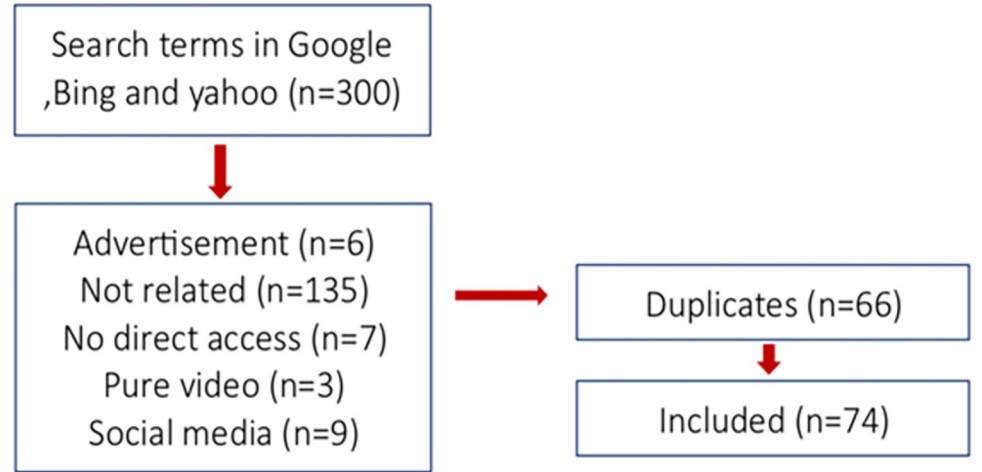

**Fig 1. Flow chart of the search strategy.**

**Table 1. Affiliation and content type and presentation Of the included websites (N = 74) data presented as frequency (N) and percentages (%).**

| Category | Criteria | N(%) |
|---|---|---|
| Affiliation | Commercial | 17 (23%) |
| | Non-profit organization | 54(73%) |
| | University/medical center | 3(4%) |
| | Governmental | 0 |
| Specialization | Exclusively related | 0 |
| | Partly related | 74(100%) |
| Content type | Medical facts | 72(95.9%) |
| | Clinical trials | 1 (1.4%) |
| | Human interest stories | 0 |
| | Question and answer | 1 (1.4%) |
| presentation | Image | 19 (25.7%) |
| | Video | 2 (2.7%) |
| | Audio | 0 |

## Quality assessment

Three different tools were used to assess the quality of the websites: the DISCERN instrument, the JAMA benchmarks, and the HONcode. Notably, none of the websites received the HONcode seal, which signifies adherence to a set of ethical and quality standards for medical information online.

The results of the DISCERN assessment are presented in Table 2, showing the median total score (sum of questions 1 to 15) as 39.5 (22). The reliability score, which includes questions 1 to 8, has a median of 23 (11). Additionally, the treatment score, comprising questions 9 to 15, has a median of 15 (6). The median and interquartile range value of each question with the maximum and minimum scores is reported in **Table 4**.

**Table 2. DISCERN evaluation for the included websites (n = 74) data presented as median with interquartile range (IQR), maximum (Max), and minimum (Min).**

| Domain | DISCERN question | Median (IQR) | Max | Min |
|---|---|---|---|---|
| Reliability | 01. Explicit aims | 3.00 (2.25) | 5 | 1 |
| | Q2. Aims achieved | 4.00 (3.25) | 5 | 1 |
| | Q3. Relevance | 3.00 (2.25) | 5 | 1 |
| | 04. Explicit sources | 1.00 (3.00) | 5 | 1 |
| | Q5. Explicit date | 5.00 (4.00) | 5 | 1 |
| | Q6. Balanced and unbiased | 5.00 (4.00) | 5 | 1 |
| | Q7. Additional sources | 1.00 (0.00) | 2 | 1 |
| | QB. Areas of uncertainty | 2.00 (1.00) | 4 | 1 |
| Treatment options | Q9. How treatment works | 2.00 (2.00) | 4 | 1 |
| | Q10. Benefits of treatment | 1.00 (0.00) | 2 | 1 |
| | Q11. Risk of treatment | 1.00 (1.00) | 4 | 1 |
| | Q12. Effects of no treatment | 2.00 (2.00) | 4 | 1 |
| | Q13. Effects on quality of life | 1.00 (1.00) | 3 | 1 |
| | Q14. All alternatives described | 3.00 (4.00) | 5 | 1 |
| | Q15. Shared decision | 1.00 (0.00) | 4 | 1 |
| Overall rating | Q16. Overall quality rating | 3.00 (1.00) | 4 | 1 |

**Table 3. Quality and readability of the included websites based on their affiliation reported as frequency and percentage (n = 74) Data presented as frequency with percentage (%).**

| Variable | Variable type | Commercial | Non-profit organization | University/medical center | Total | P-value |
|---|---|---|---|---|---|---|
| Number of achieved JAMA items per website | none | 12(16.21%) | 12(16.21%) | 1(1.35%) | 25(33.78%) | 0.002* |
| | One | 3(4.05%) | 5(6.75%) | 1(1.35%) | 9(12.16%) | |
| | Two | 1(1.35%) | 17(22.97%) | 1(1.35%) | 19(25.78%) | |
| | Three | 1(1.35%) | 20(27.02%) | 0(0%) | 21(28.38%) | |
| JAMA items | Authorship | 1(1.35%) | 36(48.64%) | 1(1.35%) | 38(51.35%) | 0.040* |
| | Attribution | 3(4.04%) | 23(31.08%) | 0(0%) | 26(19.24%) | 0.000* |
| | Currency | 4(5.40%) | 39(52.7%) | 2(2.70%) | 45(64.86%) | 0.002* |
| | Disclosure | 0(0%) | 1(0.74%) | 0(0%) | 1(0.74%) | 0.728 |
| DISCERN | Low | 17(22.97) | 49(66.41) | 3(4.04%) | 69(93.34%) | 0.194 |
| | Medium | 0(0%) | 5(6.75%) | 0(0%) | 5(6.75%) | |
| | High | 0(0%) | 0(0%) | 0(0%) | 0(0%) | |
| FRES | Readable | 15(20.70%) | 32(50%) | 2(2.70%) | 49(66.21%) | 0.063 |
| | Difficult | 2(2.70%) | 22(50%) | 1(1.35%) | 25(33.78%) | |
| FKGL | Readable | 15(20.70%) | 37(50%) | 2(2.70%) | 54(74.97%) | 0.228 |
| | Difficult | 2(2.70%) | 17(22.9%) | 1(1.35%) | 20(27.02%) | |

NA, Not Applicable

* Significant at level 0.05 or less; JAMA: Journal of American Medical Association; FRES: Flesch-Reading Ease Score; FKGL: Flesch-Kincaid Grade Level.

According to the results, Questions five and six had the highest median scores among all the questions assessed. These questions received relatively higher ratings based on the assessment criteria. On the other hand, the lowest scores were reported for Questions ten and seven, indicating that these questions received comparatively lower ratings. **Table 3** provides a summary of the median and interquartile range (IQR) of the total, reliability, treatment, and readability scores of the websites categorized by their affiliation.

According to the JAMA benchmarks, there were four key factors evaluated: authorship, attribution, disclosure, and currency. Among these, currency was the most commonly achieved aspect, observed in 45 websites (64.86%). Authorship was the second most achieved, found in 38 websites (51.35%), followed by attribution in 26 websites (19.24%). On the other hand, disclosure was the least reported factor, mentioned in only one website (0.74%). The distribution of the achieved JAMA items varied significantly based on website affiliation, as indicated in **Table 3.**

None of the websites managed to accomplish all four JAMA items simultaneously. Furthermore, in 25 websites (33.78%), none of the items were achieved. The distribution of the number of achieved items per website is presented in Table 3. Statistical analysis revealed significant variations in the distribution of achieved items per website based on the affiliation of the websites, with a p-value of less than 0.001.

## Readability assessment

The Flesch Reading Ease Score (FRES) ranged from a maximum of 100 to a minimum of 42.12, with a median of 94.75 (18.1). For the Flesch-Kincaid Grade Level (FKGL) test, the maximum score was 27.06, and the minimum score was 0.08. The median FKGL score was 5.25 (6.79), The Simple Measure of Gobbledygook (SMOG) test indicated that all websites were readable, with a median score of 3.2 (4). The maximum SMOG score was 4, and the minimum was 3.2. as detailed in Table 4.

In terms of word count, the maximum number of words on a website was 2641, while the minimum was 189, with a median of 1014.5 (706.75). Similarly, the sentence count ranged from a maximum of 310 to a minimum of five sentences, with a median of 37.5 (35.75).

**Table 4. Displays the descriptive statistics, including the minimum, maximum, median and interquartile range (IQR) values, for the variables analyzed.**

| Variable | Minimum | Maximum | Median | Interquartile Range (IQR) |
|---|---|---|---|---|
| Overall | 16 | 59 | 39.5 | 22 |
| Reliability | 8 | 35 | 23 | 11 |
| Treatment | 8 | 24 | 15 | 6 |
| FRES | 42.13 | 100 | 94.75 | 18.1 |
| FKGL | 0.08 | 27.06 | 5.25 | 6.79 |
| SMOG | 3.2 | 4 | 4 | 0.01 |
| Words | 189 | 2641 | 1014.5 | 706.75 |
| Sentences | 5 | 310 | 37.5 | 35.75 |

## Discussion

The increasing demand for orthodontic treatment highlights the need to address pain management effectively. Given that pain is a major barrier to treatment seeking, finding ways to reduce discomfort during orthodontic procedures is crucial for both patients and practitioners [3, 4]. Addressing pain in orthodontic treatment is crucial for several reasons. First, pain can negatively impact a patient's experience, leading to anxiety and reduced compliance. It can also be a sign of other problems that need attention. Managing pain promptly helps patients feel comfortable, improves treatment results, and ensures satisfaction with orthodontic care [4, 18]. With the internet's widespread distribution, people are increasingly turning to online sources for educating themselves with information about their health concerns and treatment options. This shift in information accessibility signifies a growing desire for patient autonomy and active participation in healthcare decisions [19, 20]. In the field of orthodontic pain, multiple studies were conducted to evaluate the quality of patient-centered online content in terms of YouTube videos [31, 32] and different website contents [19]. The aim of this study is to analyze the accessibility and reliability of Arabic online resources that provide patient-centered information on orthodontic pain. Given the widespread use of orthodontic treatment.

In a similar study conducted by Livas et al., they assessed the quality of English-language websites providing information about the evaluation of available internet resources regarding pain during orthodontic treatment. An analysis was conducted on 25 web pages to evaluate their accuracy, readability, accessibility, and usability. These aspects were compared to an analysis of 74 Arabic web pages from this study that used similar exclusion criteria. In the study with 25 English web pages, most sites were excluded due to the presence of links to scientific articles, as compared to this study where most exclusions were due to the presence of advertisement banners and pop-ups on Arabic content websites. Both studies aimed to evaluate the quality of web pages but had some differences in the reason why sites were filtered out of the analysis [19]. When making comparisons of the FRE ratings, the scores ranged from 48.6 to 84.7 for the English websites. In contrast, the FRE ratings for the Arabic websites spanned from 42.12 to 100.

Most of the websites incorporated into the assessment exhibited a lower FKGL score, suggesting the content was readily understandable for the average internet consumer. This conclusion was further validated by the SMOG results, confirming all of the aforementioned websites could be grasped without difficulty by someone with a middle school level of academic achievement. Furthermore, the elevated FRE score signifies the websites were indeed straightforward to comprehend. These findings align with other previous studies evaluating Arabic content, such as one that assessed orthodontic clear aligner web-based knowledge [33]. The findings together indicate that the information provided on the websites was presented in a clear way that was easy for most people to understand.

While Arabic online information about orthodontic pain aimed at patients is understandable, its reliability is lacking. Therefore, there is an urgent need to improve the quality of this type of medical content. Websites should follow accepted health information guidelines and evaluate their materials using different assessment methods to guarantee accuracy. The goal of this study is to enhance the availability and clarity of health-related resources, empowering people to make educated choices about their well-being. By informing applicable interested parties of this research, we want to promote a culture of health learning and consciousness and eventually advance health material for Arabic speakers.

Having access to orthodontic pain information that is regulated, easy to use and trustworthy could potentially provide various advantages for the general public. By obtaining more understanding about the fundamental reasons for pain, how it affects daily routines, and relief options, patients may become more adherent to treatment plans while also participating more actively in a collaborative model of decision making together with practitioners, keeping in step with international trends in healthcare delivery. Steady growth of online orthodontic informational resources and social communities will help further enhance the standard of medical care going forward. In the interim, orthodontists ought to address the current imbalance by guiding patients to education materials found on the Internet which are supported by scientific evidence.

## Limitations and recommendations

Acknowledging the inherent limitations of our study is crucial. Firstly, our analysis did not encompass social media tweets and YouTube videos, which could provide additional perspectives. Secondly, the readability tests utilized were not tailor-made for assessing Arabic text, although their relevance remains plausible. Future research should prioritize a more inclusive approach to address these limitations comprehensively. Lastly, we recommend assessors to scrutinize both Arabic and English websites for a more nuanced and accurate comparative analysis.

## Conclusion

This study assessed the quality and readability of Arabic language websites providing information on orthodontic pain. Seventy-four websites meeting the selection criteria were analyzed using tools like the DISCERN instrument, JAMA benchmarks, and readability tests. The results showed that while the text was generally understandable at a middle school reading level, the websites lacked reliability. Very few sites met the JAMA quality standards around authorship, attribution, disclosure, and currency of information. None received Health on the Net accreditation. Most were run by non-profit organizations and provided basic medical facts but lacked other important content types like clinical trials. This highlights the need to improve the quality and standards of online Arabic orthodontic pain resources. Reliable information is important to help patients make informed treatment decisions and manage their pain. Future studies should conduct a more comprehensive analysis, including social media and videos, as well as comparing Arabic and English language sites. This could provide valuable insights towards enhancing online health information for Arabic speakers.

## Supporting information

**S1 Dataset. List on included links.**
(XLSX)

## Author Contributions

**Conceptualization:** Aseel M. Aloufi, Ahmed S. Khoshhal.

**Data curation:** Muath Saad Alassaf, Hatem Hazzaa Hamadallah, Aseel M. Aloufi, Khalid N. Al-Turki, Rawah Eshky.

**Formal analysis:** Hatem Hazzaa Hamadallah.

**Funding acquisition:** Muath Saad Alassaf.

**Investigation:** Khalid N. Al-Turki.

**Methodology:** Abdulrahman Almuzaini, Khalid N. Al-Turki.

**Project administration:** Muath Saad Alassaf, Abdulrahman Almuzaini.

**Supervision:** Muath Saad Alassaf, Ahmed S. Khoshhal, Mahmoud A. Alsulaimani, Rawah Eshky.

**Validation:** Muath Saad Alassaf, Ahmed S. Khoshhal, Mahmoud A. Alsulaimani.

**Writing – original draft:** Muath Saad Alassaf, Abdulrahman Almuzaini, Aseel M. Aloufi, Ahmed S. Khoshhal, Mahmoud A. Alsulaimani, Rawah Eshky.

**Writing – review & editing:** Muath Saad Alassaf, Ahmed S. Khoshhal, Mahmoud A. Alsulaimani, Rawah Eshky.

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
