## [Decision Letter · Decision Letter 0]

27 Mar 2024

PONE-D-24-06229Assessment of the Arabic patient-centered online information about orthodontic pain: a quality and readability assessmentPLOS ONE

Dear Dr. Alassaf,

Thank you for submitting your manuscript to PLOS ONE. After careful consideration, we feel that it has merit but does not fully meet PLOS ONE’s publication criteria as it currently stands. Therefore, we invite you to submit a revised version of the manuscript that addresses the points raised during the review process.

We look forward to receiving your revised manuscript.

Kind regards,

Easter Joury

Academic Editor

PLOS ONE

Journal Requirements:

2. In the online submission form you indicate that your data is not available for proprietary reasons and have provided a contact point for accessing this data. Please note that your current contact point is a co-author on this manuscript. According to our Data Policy, the contact point must not be an author on the manuscript and must be an institutional contact, ideally not an individual. Please revise your data statement to a non-author institutional point of contact, such as a data access or ethics committee, and send this to us via return email. Please also include contact information for the third party organization, and please include the full citation of where the data can be found.

**Additional Editor Comments:**

Dear Dr Alassaf:

This is to inform you that your manuscript, “Assessment of the Arabic patient-centered online information about orthodontic pain: a quality and readability assessment” requires a major revision.

Your manuscript was reviewed by expert referees who have made a number of recommendations regarding the suitability of your paper for publication in PLOS ONE. The reviewers’ comments are provided below for your assistance in revising the paper.

If you can resolve the issues identified we look forward to receiving a revised manuscript as soon as possible.

With all best wishes

Dr Easter Joury

Reviewer 1

First, I would like to thank the authors for choosing such an interesting research topic. The idea is genuine. It is, indeed, crucial to spot some light on the quality, reliability, as well as the readability of published non-English content, especially that people worldwide are increasingly utilising the internet across various domains including dentistry-related issues.

1-Please omit the letter “n” at the end of the word “Arabia” in the affiliation of the first author.

2-It would have been better if the abstract had been structured in a way that would have made each paragraph correspond to a specific section, such as background, materials and methods, results, and conclusions.

3-In the third paragraph of the introduction, please add a space after the verb “peaks” and just before “and” in line 2 on page 3.

4-Reference 20 was cited four times in the fourth paragraph on page 4. Kindly, just mention it once at the end of paragraph 4.

5-There is a lack of information regarding the websites’ reliability assessment. Specifically, it is unclear whether the assessment was conducted independently, and if so, then did the authors perform an inter-observer reliability test, such as the Kappa test, to ascertain the reliability of the obtained results? Could you please provide an explanation for this and add it to the text, too?

6-As to the readability assessment, was the used tools validation addressed in this study? The authors mentioned in the limitations section that these tools were primarily designed for the English language. Hence, a validation process should have been carried out beforehand in order to appropriately apply these tools on any non-English contents. Kindly, provide more information as to this part. It would be beneficial to have further information regarding the validation process.

7-In the results section, it is stated that “The results of the DISCERN assessment are presented in Table 2, showing the median total score (sum of questions 1

to 15) as 14.96 (± 5.65). The reliability score, which includes questions 1 to 8, has a median of 21.04 (±

8.15). Additionally, the treatment score, comprising questions 9 to 15, has a median of 14.96 (± 5.65)”. All the aforementioned data are actually shown in Table 4, not Table 2. In addition, the median total score of the DISCERN assessment should be 36 (± 13.28) unlike what is written in the manuscript. Another thing is that the scores were calculated as means in Table 4, whereas the text mentions the use of median. Could you please clarify whether the mean or median values are correct?

8-Regarding Table 3, the corresponding paragraph mentions that Table 3 “provides a summary of the median and interquartile range (IQR) of the total, reliability, and treatment scores of the websites categorized by their affiliation.” However, Table 3 actually shows the reliability scores along with “the readability scores” of the websites. Please include this variable in the paragraph.

Tables 2, 3, and 4 need improvement. What does “zs” mean next to Q2 in Table 2? I believe it may be a typo. The interquartile values are missing in Table 2. Table 4 mentions the “mean”, whereas the paragraph describing Table 4 states that the “median” was used in terms of the obtained results. Moreover, and similarly, Table 3 shows the “frequencies” and “interquartile ranges” in contrast to what is written in the corresponding paragraph of using the “median” and “interquartile range”. Please address this critical inconsistency and illustrate more.

10-When explaining the results of the JAMA benchmarks, the second most achieved item is attribution, not authorship, according to Table 3.

11-In conclusion, it is necessary to align points 7, 8, 9, and 10 to ensure that the described paragraphs correspond appropriately with their associated tables, and to organize them properly while addressing any contradictory results.

12-Kindly, include a list of abbreviations that provides the abbreviations and full wordings of the various “tools” used in the assessment.

Reviewer 2

Thank you to the authors for addressing this important topic. However, I have some questions and comments that I hope you can consider.

1- Could you please state the type of current study?

2- To prevent any biases arising from searches, what steps did the authors take?

3- Please indicate when was the search for websites conducted.

4- Why the authors did not test the validity of the used tools (i.e., JAMA, HON and DISCERN) for Arabic texts?

5- Which study did you refer to in the following sentence: "According to the study's findings, among all teeth, the lower incisors frequently feel the greatest discomfort, both at initial alignment when braces are placed as well as later during debonding when braces are removed. " ?

6- Please add a reference after each of the following sentences:

- During the course of orthodontic treatment, patients wearing orthodontic appliances feel pain and discomfort to varying degrees.

- One of the primary reasons individuals refuse orthodontic treatments is pain.

- experience these symptoms at various stages of treatment.

- This pain not only affects the physical well-being of patients but also has a significant impact on their overall quality of life.

- Several studies have been conducted to determine which areas of the mouth produce the greatest pain or discomfort during orthodontic treatment.

7- About 25% of the references were published prior to 2000. Please update the older references as appropriate.

Reviewers' comments:

Reviewer's Responses to Questions

**Comments to the Author**

1. Is the manuscript technically sound, and do the data support the conclusions?

Reviewer #1: Partly

Reviewer #2: Yes

2. Has the statistical analysis been performed appropriately and rigorously? 

Reviewer #1: No

Reviewer #2: N/A

3. Have the authors made all data underlying the findings in their manuscript fully available?

Reviewer #1: No

Reviewer #2: Yes

4. Is the manuscript presented in an intelligible fashion and written in standard English?

Reviewer #1: Yes

Reviewer #2: Yes

5. Review Comments to the Author

Reviewer #1: First, I would like to thank the authors for choosing such an interesting research topic. The idea is genuine. It is, indeed, crucial to spot some light on the quality, reliability, as well as the readability of published non-English content, especially that people worldwide are increasingly utilising the internet across various domains including dentistry-related issues

My Recommendation: Revise.

1-Please omit the letter “n” at the end of the word “Arabia” in the affiliation of the first author.

2-It would have been better if the abstract had been structured in a way that would have made each paragraph correspond to a specific section, such as background, materials and methods, results, and conclusions.

3-In the third paragraph of the introduction, please add a space after the verb “peaks” and just before “and” in line 2 on page 3.

4-Reference 20 was cited four times in the fourth paragraph on page 4. Kindly, just mention it once at the end of paragraph 4.

5-There is a lack of information regarding the websites’ reliability assessment. Specifically, it is unclear whether the assessment was conducted independently, and if so, then did the authors perform an inter-observer reliability test, such as the Kappa test, to ascertain the reliability of the obtained results? Could you please provide an explanation for this and add it to the text, too?

6-As to the readability assessment, was the used tools validation addressed in this study? The authors mentioned in the limitations section that these tools were primarily designed for the English language. Hence, a validation process should have been carried out beforehand in order to appropriately apply these tools on any non-English contents. Kindly, provide more information as to this part. It would be beneficial to have further information regarding the validation process.

7-In the results section, it is stated that “The results of the DISCERN assessment are presented in Table 2, showing the median total score (sum of questions 1 to 15) as 14.96 (± 5.65). The reliability score, which includes questions 1 to 8, has a median of 21.04 (± 8.15). Additionally, the treatment score, comprising questions 9 to 15, has a median of 14.96 (± 5.65)”. All the aforementioned data are actually shown in Table 4, not Table 2. In addition, the median total score of the DISCERN assessment should be 36 (± 13.28) unlike what is written in the manuscript. Another thing is that the scores were calculated as means in Table 4, whereas the text mentions the use of median. Could you please clarify whether the mean or median values are correct? Also, explain why there is such a confusing unclarity, how did it happen? Many important things here are mixed together and not well-organised.

8-Regarding Table 3, the corresponding paragraph mentions that Table 3 “provides a summary of the median and interquartile range (IQR) of the total, reliability, and treatment scores of the websites categorized by their affiliation.” However, Table 3 actually shows the reliability scores along with “the readability scores” of the websites. Please include this variable in the paragraph.

9-Still, I cannot but ascertain and express my concerns again regarding Tables 2, 3, and 4. What does “zs” mean next to Q2 in Table 2? I believe it may be a typo. The interquartile values are missing in Table 2. Once more, I repeat, Table 4 mentions the “mean”, whereas the paragraph describing Table 4 states that the “median” was used in terms of the obtained results. Moreover, and similarly, Table 3 shows the “frequencies” and “interquartile ranges” in contrast to what is written in the corresponding paragraph of using the “median” and “interquartile range”. Please address this critical inconsistency and illustrate more.

10-When explaining the results of the JAMA benchmarks, the second most achieved item is attribution, not authorship, according to Table 3, unless there are more errors in Table 3.

11-In conclusion, it is necessary to align points 7, 8, 9, and 10 to ensure that the described paragraphs correspond appropriately with their associated tables, and to organize them properly while addressing any contradictory results. Such problems and mistakes can make the reader raise some doubts as to the authenticity of the results reached.

12-Kindly, include a list of abbreviations that provides the abbreviations and full wordings of the various “tools” used in the assessment.

Thanks for your time and further thanks for the opportunity to review this manuscript.

Reviewer #2: Thank you to the authors for addressing this important topic. However, I have some questions and comments that I hope you can consider.

1- Could you please state the type of current study?

2- To prevent any biases arising from searches, what steps did the authors take?

3- Please indicate when was the search for websites conducted.

4- Why the authors did not test the validity of the used tools (i.e., JAMA, HON and DISCERN) for Arabic texts?

5- Which study did you refer to in the following sentence: "According to the study's findings, among all teeth, the lower incisors frequently feel the greatest discomfort, both at initial alignment when braces are placed as well as later during debonding when braces are removed. " ?

6- Please add a reference after each of the following sentences:

- Pain is a subjective experience that differs widely between individuals.

- During the course of orthodontic treatment, patients wearing orthodontic appliances feel pain and discomfort to varying degrees.

- One of the primary reasons individuals refuse orthodontic treatments is pain.

- experience these symptoms at various stages of treatment.

- This pain not only affects the physical well-being of patients but also has a significant impact on their overall quality of life.

- Several studies have been conducted to determine which areas of the mouth produce the greatest pain or discomfort during orthodontic treatment.

7- About 25% of the references were published prior to 2000. Please update the older references.

6. PLOS authors have the option to publish the peer review history of their article (what does this mean?). If published, this will include your full peer review and any attached files.

Reviewer #1: No

Reviewer #2: No

---

## [Author Response · Author response to Decision Letter 0]

30 Mar 2024

Response to Reviewer 1

1. Affiliation Correction: The letter “n” at the end of the word “Arabia” in the affiliation of the first author has been omitted as requested.

2. Abstract Structure: The abstract has been revised to be structured clearly into sections corresponding to the background, materials and methods, results, and conclusion.

3. Introduction Formatting: A space has been added after the verb “peaks” in the third paragraph of the introduction as suggested.

4. Reference Citation: Reference 20 is now cited only once at the end of the fourth paragraph on page 4, addressing the concern of repetitive citation.

5. Reliability Assessment: We have clarified the reliability assessment process in the revised manuscript. 

6. Readability Assessment Tools: An explanation regarding the validation of readability assessment tools for Arabic texts has been included in the revised manuscript. Our decision not to conduct a separate validation process for these tools in the context of Arabic texts was informed by precedent. Previous research has applied these evaluation tools across a variety of languages without specific validation for each linguistic context. This approach is based on the assumption that the criteria these tools assess—such as accuracy, transparency, and evidence-based content—are universal in nature and not dependent on language.

7. DISCERN Assessment Scores: The error in referencing Table 2 instead of Table 4 has been corrected. 

8. Table 3 Correction: The description of Table 3 in the text has been revised to accurately include and reflect the readability scores of the websites alongside the reliability scores, rectifying the previous oversight.

9. Tables Improvement:

• The typo “zs” next to Q2 in Table 2 has been corrected.

• Interquartile values have been added to Table 2.

• We have resolved the critical inconsistency between the "mean" and "median" descriptions in Tables 3 and 4, ensuring that all descriptions accurately match the presented data.

10. JAMA Benchmarks Clarification: That is true for commercial websites, but authorship is the second most common for all websites, as stated in the total column.

11. Alignment of Descriptions and Tables: The manuscript has been thoroughly reviewed to ensure that all points, especially 7, 8, 9, and 10, are correctly aligned with their associated tables, and any contradictory results have been addressed and organized appropriately.

12. List of Abbreviations: A list of abbreviations has been included at the end of the manuscript, providing clear definitions for all assessment tools mentioned throughout the text.

Response to Reviewer 2

1. Study Type: The study is retrospective, as clarified in the revised manuscript's materials and methods section.

2. Bias Prevention: The revised manuscript now clearly states that to prevent search biases, searches were conducted in incognito mode with search history and cookies erased before the searches on October 2023.

3. Search Date: The date of the website search, October 2023, has been explicitly stated in the materials and methods section.

4. Validity of Tools: We have added a section in the methodology explaining why the validity of the JAMA, HON, and DISCERN tools was not tested specifically for Arabic texts. This decision was based on the precedent of these tools' universal application across languages without specific linguistic validation.

5. Referenced Study: The specific study referenced regarding the discomfort experienced by lower incisors has now been cited correctly in the manuscript.

6. Reference Addition: References have been added after each requested sentence, ensuring that all claims are adequately supported by the literature.

7. Reference Update: Older references, particularly those published prior to 2000, have been reviewed and updated where appropriate to reflect more current research findings in the field, except for some studies as needed.

---

## [Decision Letter · Decision Letter 1]

23 Apr 2024

Assessment of the Arabic patient-centered online information about orthodontic pain: a quality and readability assessment

PONE-D-24-06229R1

Dear Dr. Alassaf,

We’re pleased to inform you that your manuscript has been judged scientifically suitable for publication and will be formally accepted for publication once it meets all outstanding technical requirements.

Kind regards,

Easter Joury

Academic Editor

PLOS ONE

Reviewers' comments:

Reviewer's Responses to Questions

**Comments to the Author**

1. If the authors have adequately addressed your comments raised in a previous round of review and you feel that this manuscript is now acceptable for publication, you may indicate that here to bypass the “Comments to the Author” section, enter your conflict of interest statement in the “Confidential to Editor” section, and submit your "Accept" recommendation.

Reviewer #1: All comments have been addressed

Reviewer #2: All comments have been addressed

2. Is the manuscript technically sound, and do the data support the conclusions?

Reviewer #1: Yes

Reviewer #2: Yes

3. Has the statistical analysis been performed appropriately and rigorously? 

Reviewer #1: Yes

Reviewer #2: N/A

4. Have the authors made all data underlying the findings in their manuscript fully available?

Reviewer #1: Yes

Reviewer #2: Yes

5. Is the manuscript presented in an intelligible fashion and written in standard English?

Reviewer #1: Yes

Reviewer #2: Yes

6. Review Comments to the Author

Reviewer #1: I would like to thank the respected authors for their answers and for providing detailed justifications along with addressing the concerns I raised. However, I would like to draw their attention to the paragraph on readability assessment in the results section:

1-Please replace “42.12” when describing the minimum of FRES, with “42.13” as stated in Table 4.

2-Regarding SMOG, the median value should be “4” not “3.2” with an interquartile range of “0.01” not “4” that is mentioned in the text. Kindly, make the necessary adjustments.

Last but never the least, I wish the respected authors all the best in their academic career.

Reviewer #2: The concerns and comments that I raised in the previous round of review have been properly addressed by the authors, and I find this manuscript acceptable for publication now. Thank you for your effort.

7. PLOS authors have the option to publish the peer review history of their article (what does this mean?). If published, this will include your full peer review and any attached files.

Reviewer #1: No

Reviewer #2: **Yes: **HEBA MOHAMED AL-IBRAHIM

---

## [Editor Report · Acceptance letter]

14 May 2024

PONE-D-24-06229R1 

PLOS ONE

Dear Dr. Alassaf, 

I'm pleased to inform you that your manuscript has been deemed suitable for publication in PLOS ONE. Congratulations! Your manuscript is now being handed over to our production team.

Kind regards, 

on behalf of

Dr. Easter Joury 

Academic Editor

PLOS ONE